# RETHINKING THE NOISE SCHEDULE OF DIFFUSION-BASED GENERATIVE MODELS

## ABSTRACT

In this work, we undertake both theoretical and empirical analysis of noise scheduling strategies within the scope of denoising diffusion generative models. We investigate the training noise schedule through the lens of power spectrum and introduce a novel metric, weighted signal-noise-ratio (WSNR), to uniformly represent the noise level in both RGB and latent spaces, enhancing the performance of high-resolution models in these spaces with WSNR-Equivalent training noise schedules. Further, we examine the reverse sampling process using the framework of Ordinary Differential Equations (ODEs), elucidating the concept of the optimal denoiser and providing insights into data-driven sampling noise schedules. We explore the correlation between the number of evaluation points and the generation quality to optimize the acceleration of the ODE solver in the diffusion model. Based on practical considerations of evaluation point effects, we propose an adaptive scheme to choose numerical methods within computational constraints, balancing efficacy and efficiency. Our approach, requiring no additional training, refines the FID of pre-trained CIFAR-10 and FFHQ-64 models from 1.92 and 2.45 to 1.89 and 2.25, respectively, utilizing 35 network evaluations per image.

## 1 INTRODUCTION

Denoising diffusion generative models (Song & Ermon, 2019; Ho et al., 2020; Song et al., 2020b; Karras et al., 2022; Meng et al., 2023; Xue et al., 2023; Song et al., 2023) have become crucial in developing state-of-the-art generative models due to their ability to generate new, unseen data samples after training on an existing dataset. They have demonstrated unprecedented success in the synthesis of text-to-image (Ramesh et al., 2022; Saharia et al., 2022; Balaji et al., 2022), 3D objects (Poole et al., 2022; Lin et al., 2022; Shue et al., 2022; Bautista et al., 2022), audio (Kong et al., 2020), time series (Tashiro et al., 2021; Biloš et al., 2022), and molecules (Wu et al., 2022; Qiao et al., 2022; Xu et al., 2022). Among these, denoising diffusion generative models have garnered considerable attention due to their exceptional ability to generate high-quality synthetic data. However, current noise scheduling strategies remains handcrafting for each dataset, such as VP (Ho et al., 2020), VE (Song et al., 2020b), Cosine (Nichol & Dhariwal, 2021) and EDM (Karras et al., 2022). These noise schedules perform well in low-resolution RGB spaces but yield poorer results in higher resolutions (Dhariwal & Nichol, 2021; Hoogeboom et al., 2023). Additionally, we observe that the sampling efficiency of diffusion models can be enhanced by employing different sampling noise schedules for various datasets. These observations lead us to rethink the noise schedule of diffusion-based generative models.

We first analyze and quantify the noise level in data during the forward process of diffusion models. As illustrated in Fig. 2, a notable variance in noise levels across images of different resolutions when subjected to Gaussian noise with identical standard deviations. Specifically, high-resolution images display lower noise levels, while lower-resolution images exhibit higher noise levels. Inspired by this observation, we conduct frequency domain analysis on power spectrum, where we notice a uniform average power spectrum of isotropic Gaussian noise across all frequencies. Delving deeper, we propose a novel metric, weighted signal-noise-ratio (WSNR), weighting SNR for each frequency component, and observe that WSNR is consistent across different resolutions at the same noise level.

For the sampling process of the diffusion model (Song et al., 2020b), an ODE solver is typically employed for simplicity. Under the framework of probability flow ODE, the diffusion model predicts a clean image, $\mathbf{x}_0$, at each step based on the current noisy data $\mathbf{x}_t$, which can be viewed as a denoiser, $D(\mathbf{x}_t)$. Given a finite dataset, an ideal solution for the denoiser $D(\mathbf{x}_t)$ can be found as the weighted sum of all clean data in the dataset. This weight is normalized by a softmax function, related to the distance between the noisy data $\mathbf{x}_t$ and each clean data point. Building on this ideal solution, we delve into the relationship between the probability distribution of the generated data and the initial noise distribution at the start point. We discover that the diversity of the generated data is jointly influenced by the initial Gaussian distribution at the start point and the Euclidean distance from the data points in the dataset (as illustrated in Fig. 4 and elaborately explained in Sec. 5). Drawing from this analysis, we propose a data-driven sampling noise schedule to determine the integration interval of the ODE for the balance of the efficiency and generation quality.

Additionally, as shown in Tab. 4, we examine the number of evaluation points, specifically the unique time steps at which the diffusion model makes predictions, and analyze its impact on the generation quality. Our study reveals that an increased number of evaluation points leads to better results when the step size is relatively large. Conversely, with smaller step sizes, the advantage of adding more evaluation points becomes less significant. Drawing on these findings, we propose a strategy for dynamically selecting numerical methods according to computational constraints, aiming to optimize generation quality. Our approach, requiring no additional training, refines the FID of pre-trained CIFAR-10 and FFHQ-64 models from 1.92 and 2.45 to 1.89 and 2.25, respectively, utilizing 35 network evaluations per image.

Our contributions are listed as follows: (1) We propose a novel metric, WSNR, to consistently quantifies the noise level of the training data in both the RGB space and latent space. A WSNR-Equivalent training noise schedule is proposed to improve the performance of diffusion model in the RGB space and latent space. (2) In theory, we analyze that the ODE sampling noise schedule should be data-driven. Consequently, we propose estimating the integration interval based on the average data distance, which achieves the trade-off between the quality of generated data and computational cost. (3) We empirically explore the relationship between the number of evaluation points and the generation quality. Our findings lead us to develop a Number of Function Evaluations (NFE)-guided sampling noise schedule, which enables dynamic switching of numerical methods based on the allocated NFE budget. (4) Our approach refines the FID of pre-trained CIFAR-10 and FFHQ-64 models from 1.92 and 2.45 to 1.89 and 2.25, respectively, utilizing 35 network evaluations per image.

## 2 RELATED WORK

Score-based diffusion models (Song et al., 2020b; Ho et al., 2020) are a generative model that perturbes data with Gaussian noise through a diffusion process for training, and the reverse process is learned to transform the Gaussian distribution back to the data distribution. Perturbing data points with noise populates low data density regions to improve the accuracy of estimated scores, resulting in stable training and image sampling. The forward process is controlled by the handcrafted noise schedule. Recent works (Chen, 2023; Hoogeboom et al., 2023) propose the carefully designed noise schedule and demonstrate the schedule is superior to the VP, VE and Cosine schedule on the RGB space. Different from the previous works, our method is based on the analysis of power spectrum and quantifies the noise level numerically. Besides, the WSNR-Equivalent noise schedule is valid on both the latent space and RGB space, which has not been discussed in the previous works. The SNR concept is first proposed in (Kingma et al., 2021). Different from SNR, WSNR is consistent across different resolutions and even in latent space.

The probability flow ODE is first introduced by (Song et al., 2020b). Under the ODE framework, DDIM (Song et al., 2020a) is identical to the explicit Euler method. Jolicoeur-Martineau et al. (2021) uses a standard higher-order adaptive SDE solver to accelerate the sampling process. (Lu et al., 2022) proposes to use the high-order ODE solver in $\log(\sigma)$ space to greatly accelerate the sampling of diffusion models while keep considerable generation quality. (Zhang & Chen, 2022) is a concurrent work with (Lu et al., 2022), which leverages a semilinear structure of the learned diffusion process to reduce the discretization error. Different from the previous works, our analysis aims at exploring the relationship between the sampling noise schedule and dataset, rather than the

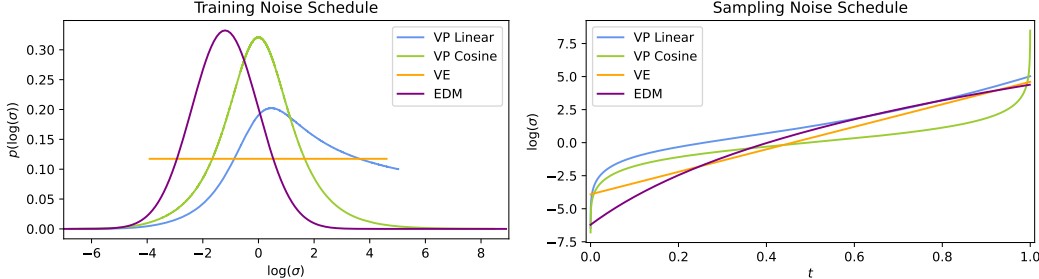

Figure 1: **Left:** Representation of the training noise schedule in the $\log(\sigma)$ space, expressed in probability density format. **Right:** The sampling noise schedule represented in the time $t$ space.

acceleration of ODE solver. The ideal denoiser is first introduced in (Karras et al., 2022), which is to simplify the design space of diffusion-based generative models. Our analysis is further extend the solution of ideal denoiser to analyze the integration interval of sampling process. Our dynamic schedule only includes the explicit Runge-Kutta methods, such as Heun's method, Midpoint method and 3rd-order method, which targets at exploring of the relationship between the number of evaluation points and the generation quality.

## 3 SIMPLE EXPRESSION OF DIFFUSION MODEL

The forward diffusion process commences with a clean image $\mathbf{y} \sim p_{\text{data}}(\mathbf{y})$. This process introduces isotropic Gaussian noise $\sigma\mathbf{n} \sim \mathcal{N}(\mathbf{0}, \sigma^2\mathbf{I})$ into the clean image $\mathbf{y}$, with the goal of populating low data density regions. This action is designed to enhance the accuracy of estimated gradient scores $\nabla_\mathbf{x}\log p(\mathbf{x})$, where $\mathbf{x} = \mathbf{y} + \sigma\mathbf{n}$. The forward diffusion process is described by a stochastic differential equation (SDE), which maintains the desired distribution $p$ as sample $\mathbf{x}$ evolves over time (Song et al., 2020b). The corresponding probability flow ordinary differential equation (ODE) enables a deterministic process whose trajectories share the same marginal probability densities. The forward SDE and reverse ODE are formulated as:

$$\begin{aligned} \text{Forward SDE:} \quad & \mathrm{d}\mathbf{x} = \sqrt{2\sigma}\mathrm{d}\mathbf{w} \\ \text{Reverse ODE:} \quad & \mathrm{d}\mathbf{x} = \frac{(\mathbf{x} - D(\mathbf{x}; \sigma))}{\sigma}\mathrm{d}\sigma, \end{aligned} \quad (1)$$

where $D(\mathbf{x}; \sigma)$ is a denoiser function. Pioneering diffusion models (Ho et al., 2020; Song et al., 2020b; Kingma et al., 2021) characterize a noise schedule, $\sigma(t)$, and sample from a distribution $t \sim \mathcal{U}(t_{min}, t_{max})$. According to "Method of Jacobians", the conventional noise schedule rooted in variable $t$ can be cast as a probability density function (PDF): $p(\sigma) = p_t(t^{-1}(\sigma))|\mathrm{d}t/\mathrm{d}\sigma(t)|$, where $p_t$ signifies the PDF associated with the time variable $t$, illustrated in Fig. 1. The sampling approach leans on the advanced ODE numerical methods, enabling discretization of the time variable $t$ at uniform intervals. As highlighted by (Karras et al., 2022), it's noteworthy that the noise schedule during sampling, $\sigma_{sample}(t)$, can differ from the training noise schedule, $\sigma(t)$.

## 4 UNDERSTANDING DIFFUSION MODEL FROM POWER SPECTRUM

Typically, the same noise schedule $\sigma(t)$ is applied across many image datasets. Nevertheless, the noise level in the noisy image $\mathbf{x}$ relative to the original clean image $\mathbf{y}$ demonstrates variability depending on the image size, as illustrated in Fig. 2. This variability arises due to the noise schedule overlooking a prior knowledge embedded in images, including valuable information from neighboring pixels. We introduce a novel metric, the Weighted Signal-to-Noise Ratio (WSNR), which facilitates a coherent portrayal of the noise level, irrespective of image resolution variations. WSNR hinges on the expected power spectrum across distinct frequency components. More precisely, the WSNR of a noisy image $\mathbf{y} + \sigma\mathbf{n}$ with dimensions $(C, H, W)$ is defined as:

$$\text{WSNR}(P_\mathbf{y}, \sigma) = \sum_{c=0}^{C-1}\sum_{u=0}^{H-1}\sum_{v=0}^{W-1} \frac{P_{\mathbf{y},c}(u,v)}{\sum_{u=0}^{H-1}\sum_{v=0}^{W-1} P_{\mathbf{y},c}(u,v)} \frac{P_{\mathbf{y},c}(u,v)}{\mathbb{E}[P_{\sigma\mathbf{n},c}(u,v)]}, \quad (2)$$

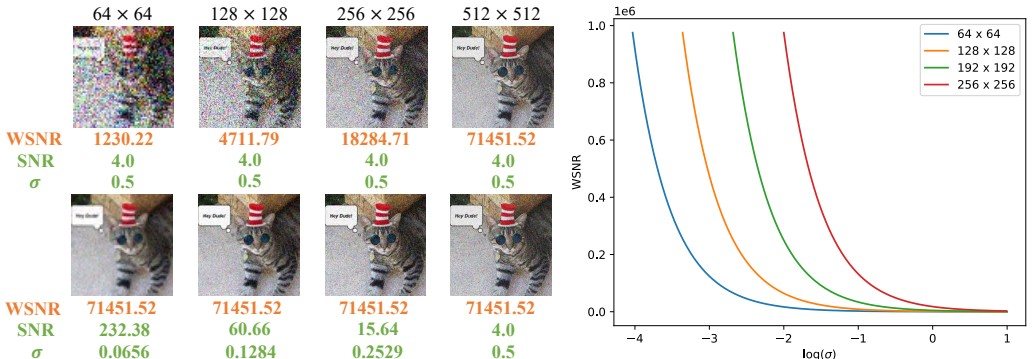

Figure 2: Comparison of noisy images with the same $\sigma$, SNR and WSNR. The top row reveals that higher-resolution images exhibit lower noise levels when additional Gaussian noise with $\sigma = 0.5$ is applied. The bottom row demonstrates similar noise levels in noisy images with the same WSNR.

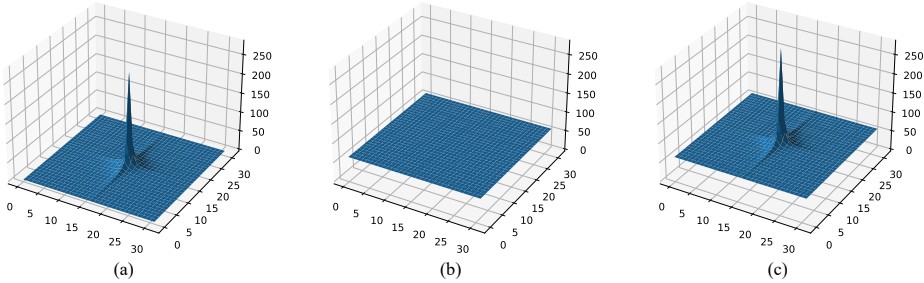

Figure 3: (a) The power spectrum of CIFAR-10 dataset, suggesting that natural scenes are dominated by low-frequency components with a sharp decline as move to high-frequency components. (b) The expected power spectrum of the isotropic Gaussian noise $\sigma \mathbf{n}$. Within this spectrum, each frequency component maintains a consistent power, $\sigma^2$. (c) The expected power spectrum of the noisy images. As the images and the Gaussian noise are independent, the expected power spectrum is simply equal to the summation of (a) and (b).

where $P_{\cdot,c}(u, v)$ is the power of the frequency component at $(u, v)$ within the $c$-th channel. The first term signifies the proportion of power contributed by each frequency component to the total power, while the second term represents the signal-to-noise ratio at that particular frequency component. For isotropic Gaussian noise $\sigma \mathbf{n}$, the expected power spectrum of each frequency component remains constant at $\sigma^2$, as illustrated in Fig. 3. This can be formulated as follows:

$$\mathbb{E}[P_{\sigma \mathbf{n}}(u, v)] = \sigma^2 \mathbb{E}[F_{\mathbf{n}}(u, v) F_{\mathbf{n}}^*(u, v)] = \frac{\sigma^2}{HW} \sum_{k=0}^{H-1} \sum_{l=0}^{W-1} \mathbb{E}[\mathbf{n}(k, l)^2] = \sigma^2. \tag{3}$$

For simplicity, the channel index $c$ is ignored in Eq. 3. See Appendix for the detailed derivation.

In the right plot of Fig. 2, a notable differentiation in the WSNR curves is evident among images with diverse resolutions. Employing a constant training noise schedule, illustrated in the left plot of Fig. 1 is inappropriate across multiple resolutions in RGB space and latent space data. We propose WSNR-Equivalent training noise schedule aimed at maintaining consistency in the PDF of WSNR, $p(\text{WSNR}(\mathbb{E}_{\mathbf{y} \sim p_{\text{data}}(\mathbf{y})}[P_{\mathbf{y}}], \sigma))$, across assorted resolutions or within latent spaces. Here, the averaged power spectrum across the entire training dataset is used as a proxy.

### 4.1 EXPERIMENTS WITH WSNR-EQUIVALENT TRAINING NOISE SCHEDULE

We elucidate the experiments conducted to validate the effectiveness of the WSNR-Equivalent Training Noise Schedule in advancing the performance of diffusion models. We show that the WSNR-Equivalent Training Noise Schedule can benefit both the generation in RGB space and latent space.

Table 1: FID on FFHQ dataset at different resolutions. $p(\text{WSNR})$ is our WSNR-Equivalent training noise schedule. $p(\sigma)$ means the EDM training noise schedule.

| Resolution | 64×64 | 128×128 | 256×256 |
|---|---|---|---|
| $p(\text{WSNR})$ | 3.70 | **6.15** | **7.89** |
| $p(\sigma)$ | 3.70 | 7.13 | 11.49 |

Table 2: FID on ImageNet 256×256 dataset. Both the models are trained in the 32×32 latent space, which show that our training noise schedule benefits the latent diffusion model.

| Network | Params(↓) | FID(↓) |
|---|---|---|
| UViT-M, $p(\text{WSNR})$ | **131M** | **3.38** |
| UViT-M, $p(\sigma)$ | **131M** | 4.80 |
| UViT-L, $p(\sigma)$ | 287M | 3.40 |

**Training details.** We first train diffusion models with WSNR-Equivalent training noise schedule across multiple scales on FFHQ dataset (Karras et al., 2019) in the RGB space and ImageNet (Deng et al., 2009) in the latent space (Rombach et al., 2022). All the training noise schedules are aligned with the $p(\text{WSNR})$ of the ImageNet dataset at 64x64 resolution in the RGB space under the EDM noise schedule (Karras et al., 2022). For a fair comparison, the models on FFHQ dataset share the same network architecture with the baseline models. For FFHQ-64×64, the network is identical to (Karras et al., 2022). For FFHQ-128×128 and FFHQ-256×256, we further increase the number of stage by 1 and 2, respectively, to ensure the resolution of *mid block* feature maps in U-Net is the same as that in FFHQ-64×64. All the models are trained for 780k iterations, with the batch size of 256 and no weight decay. The learning rate is set as 0.002 with 40000 iteration for linear warmup. The exponential moving average rate is 0.9996.

For ImageNet dataset experiment, UViT-M (Bao et al., 2023) is adopted as backbone for efficient training. We keep the same hyperparameter with the original implementation (Bao et al., 2023). The training objective function is replaced with the EDM precondition (Karras et al., 2022). We follow latent diffusion models (Rombach et al., 2022) to convert images in 256×256×3 shape to latent representations at 32×32×4 shape, using the pre-trained image autoencoder provided by (Rombach et al., 2022).

As shown in Tab. 1, there is a parity between the FID scores obtained under both noise schedules, with each exhibiting a score of 3.70, for the 64×64 resolution. However, as the resolution increases, a noticeable divergence in the FID scores is observed between the two noise schedules. In the case of 128×128 resolution, the WSNR-Equivalent noise schedule manifests a lower FID score of 6.15, implying superior image quality and diversity compared to the EDM schedule, which posts a score of 7.13. The divergence is more pronounced at the 256×256 resolution, where the FID score under the WSNR-Equivalent schedule is 7.89, significantly better than the FID score of 11.49 under the EDM noise schedule. As delineated in Tab. 2, UViT-M, leveraging our WSNR-Equivalent noise schedule, attains a superior FID score compared to the UViT-L model, whose parameter count is more than double that of UViT-M.

The EDM noise schedule is proposed to improve the training efforts at the intermediate noise levels at the low resolution, which is depicted by the standard deviation $\sigma$ of the additional Gaussian noise. The results in Tab. 1 demonstrate that our proposed WSNR is a better metric to quantize the noise level in the forward diffusion process. Tab. 2 manifests that WSNR serves as a valid metric to illustrate the noise level in the latent space.

## 5 UNDERSTANDING DIFFUSION MODEL FROM ODE PROBABILITY FLOW

Suppose that the noisy image $\mathbf{x}_i = \mathbf{y}_i + \sigma\mathbf{n}$ and the distance between clean images $\mathbf{d}_{ij} = \mathbf{y}_i - \mathbf{y}_j$, then we can derive the ideal output of the denoiser as following:

$$D(\mathbf{x};\sigma) = \frac{\sum_j \mathcal{N}(\mathbf{x};\mathbf{y}_j,\sigma^2\mathbf{I})\mathbf{y}_j}{\sum_j \mathcal{N}(\mathbf{x};\mathbf{y}_j,\sigma^2\mathbf{I})} = \sum_j \text{softmax}(\frac{||\mathbf{d}_{ij}+\sigma\mathbf{n}||^2}{-2\sigma^2})\mathbf{y}_j = \sum_j p_j\mathbf{y}_j \qquad (4)$$

The result shows that the ideal output is the weighted average of clean image and the weights are the normalized by the softmax function. The weight $p_i$ can be further derived as:

$$p_i = \frac{\exp(\frac{||\mathbf{d}_{ii}||^2+2\sigma\langle\mathbf{d}_{ii},\mathbf{n}\rangle+||\sigma\mathbf{n}||^2}{-2\sigma^2})}{\sum_j \exp(\frac{||\mathbf{d}_{ij}||^2+2\sigma\langle\mathbf{d}_{ij},\mathbf{n}\rangle+||\sigma\mathbf{n}||^2}{-2\sigma^2})} = \frac{1}{1+\sum_{j\ j\neq i}\exp(\frac{||\mathbf{d}_{ij}||^2+2\sigma\langle\mathbf{d}_{ij},\mathbf{n}\rangle}{-2\sigma^2})}, \qquad (5)$$

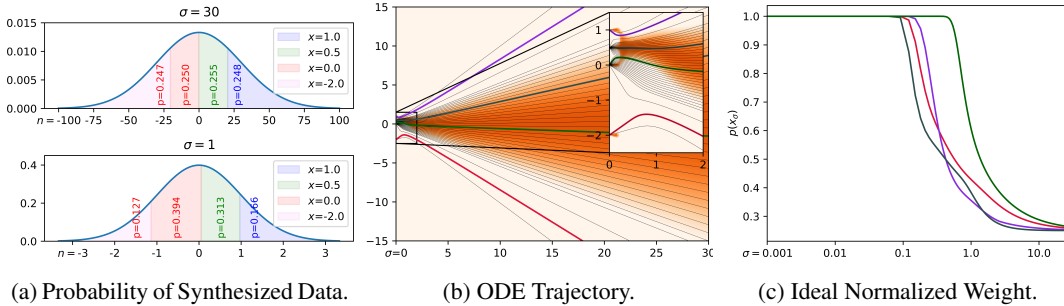

(a) Probability of Synthesized Data.     (b) ODE Trajectory.     (c) Ideal Normalized Weight.

Figure 4: Analysis of ODE Probability Flow on a toy 1D dataset, where $p_{data}$ is four Dirac peaks at $x = 1, 0.5, 0, -2$. (a) The probability of the synthesized data from Gaussian noise. As the $\sigma$ increases, the distribution of the generated data progressively converges to the true underlying distribution. (b) A sketch of ODE curvature in 1D, where the color of the bolded curves corresponds to the color of the curves in (c). (c) The softmax weight of the source data point. As shown in Eq. 4, the ideal denoiser outputs the weighted average of the clean dataset. The softmax weight approaches $\frac{1}{N}$ as $\sigma$ increases. At low noise level, the ideal prediction is the clean source data.

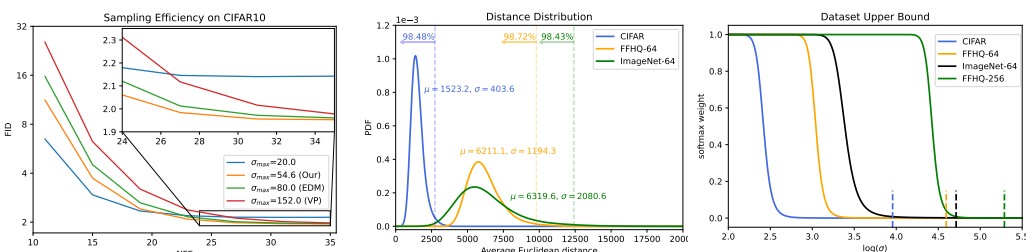

Figure 5: FID of different initial distributions on CIFAR10 using 2nd-order Heun's method. A smaller $\sigma_{max}$ yields better FID with equal NFE because of lesser step size. With higher NFE, our $\sigma_{max}$ effectively balances NFE and FID trade-off.

Figure 6: **Left:** The distribution of the average Euclidean distance $\overline{||\mathbf{d}_\star||^2}$ in Eq. 6 on CIFAR-10, FFHQ-64 and ImageNet-64 dataset. The upper-bound proxies for all datasets encompass over 98% of the samples, exceeding the theoretical upper bounds provided by Eq. 7. **Right:** The softmax weight curve of upper-bound proxies. The x-axis represents $\log(\sigma)$. At the proxy point, the softmax weight is small enough to achieve the trade-off between the image quality and sampling efficiency.

where $\mathbf{d}_{ii} = \mathbf{0}$ and $\langle \mathbf{d}_{ij}, \mathbf{n} \rangle$ represents the dot product between the distance $\mathbf{d}_{ij}$ and the standard Gaussian noise $\mathbf{n}$.

**Probability of the synthesized data.** In the reverse ODE, the source of randomness is the initial point, following a Gaussian distribution. The probability of the synthesized data depends on the solution trajectory and standard deviation of the initial distribution. In the context of data generation, the desired outcome is often a broad variety in the generated samples, implying that the distribution of these samples should approximate a uniform distribution. We observed that as the normalized weight $p_i$ approaches $\frac{1}{N}$, where $N$ is the size of dataset, the data generated from Gaussian noise tends to have a nearly uniform distribution, the actual data distribution $p_{data}$, as demonstrated in Fig. 4a. Fig. 4b and 4c indicate that the normalized weight $p_i$ approaches $\frac{1}{N}$ as $\sigma$ increases.

**Data-driven sampling noise schedule.** To ensure the diversity of generated data, the initial distribution is required to have a large standard deviation, denoted by $\sigma_{max}$, implying a lengthy integration range for ODE. In practice, while the solution of ODEs is difficult to ascertain analytically, we resort to numerical methods for approximation. However, the ODE solver introduces truncation error which is accumulated throughout the integration interval, spanning from $\sigma_{max}$ to 0. Typically, the number of step is proportional to the integration range to achieve the same accuracy, which is demonstrated by the results in Fig. 5. To achieve the same FID score on CIFAR-10 dataset, the larger

Table 3: FID score across CIFAR-10, FFHQ-64, and ImageNet-64 datasets under varied initial distributions, evaluated using three distinct ODE solvers: 2nd-order Heun's method, 3rd-order DPM Solver and our 3rd-order solver.

| Methods | CIFAR-10 (NFE=35) | | | FFHQ-64 (NFE=35) | | | ImageNet-64 (NFE=79) | | |
|---|---|---|---|---|---|---|---|---|---|
| | Heun | DPM-Solver | Our 3rd | Heun | DPM-Solver | Our 3rd | Heun | DPM-Solver | Our 3rd |
| Our $\sigma_{max}$ | 1.95 | 1.99 | **1.89** | 2.42 | 2.45 | **2.25** | 2.30 | 2.27 | **2.25** |
| EDM $\sigma_{max}$ | 1.97 | 2.00 | 1.97 | 2.43 | 2.49 | 2.30 | 2.36 | 2.33 | 2.31 |
| VP $\sigma_{max}$ | 1.98 | 2.05 | 2.00 | 2.45 | 2.50 | 2.31 | 2.35 | 2.29 | 2.28 |

Table 4: FID and NEP (Number of Evaluation Point) on CIFAR-10 dataset. In midpoint method, an additional predictive point is inserted in the middle of each interval, allowing it to outperform the second-order Heun's method when the step size is large. However, as the step size decreases, the performance of Heun's method proves superior.

| Steps | 4 | 6 | 8 | 10 | 12 | 14 | 16 | 18 |
|---|---|---|---|---|---|---|---|---|
| Midpoint FID | 22.44 | 3.85 | 2.30 | 2.12 | 2.06 | 2.04 | 2.03 | 2.02 |
| Midpoint NEP | 7 | 11 | 15 | 19 | 23 | 27 | 31 | 35 |
| Heun FID | 80.2 | 11.3 | 3.73 | 2.41 | 2.08 | 1.98 | 1.96 | 1.95 |
| Midpoint NEP | 4 | 6 | 8 | 10 | 12 | 14 | 16 | 18 |
| 3rd-Order FID | 147.3 | 22.6 | 3.31 | 2.03 | 1.90 | 1.89 | 1.90 | 1.89 |
| 3rd-Order NEP | 7 | 11 | 15 | 19 | 23 | 27 | 31 | 35 |

Figure 7: FID on CIFAR-10 of dynamic numerical method and NFE-guided sampling noise schedule. Our dynamic scheme outperforms the classical ODE methods.

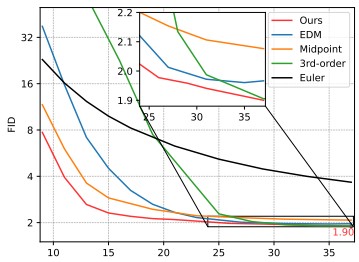

integration interval requires more neural function evaluations (NFE). For the trade-off between the quality of generated data and NFE, it's crucial to identify an appropriate value of $\sigma_{max}$ such that the normalized weight $p_i$ approaches $\frac{1}{N}$.

Based on Eq. 5, the normalized weight $p_i$ depends on the set $\{\mathbf{d}_{ij}|j \in \{0, 1, ..., N-1\} \wedge j \neq i\}$. The real-world data distance relationship is intricate and poses analytical challenges. Given that the function $f(x) = \exp(-x)$ is convex and nonnegative, we can determine the upper bound of the normalized weight $p_i$ via Jensen's inequality, which can be formulated as:

$$p_i \leq \frac{1}{1 + (N-1)\exp\left(\frac{\overline{||\mathbf{d}_i||^2} + 2\sigma\langle\overline{\mathbf{d}_i}, \mathbf{n}\rangle}{-2\sigma^2}\right)}, \tag{6}$$

where the $\overline{||\mathbf{d}_i||^2} = \frac{1}{N-1}\sum_{j\ j\neq i}||\mathbf{d}_{ij}||^2$ and $\overline{\mathbf{d}_i} = \frac{1}{N-1}\sum_{j\ j\neq i}\mathbf{d}_{ij}$. Given the dataset where each data point has a distance measure $\overline{||\mathbf{d}_i||^2}$, we can utilize Chebyshev's inequality to determine an upper bound proxy that encompasses the majority of the points in the dataset. Specifically, we designate the upper bound proxy for the entire dataset as: $||\mathbf{d}||^2 = \mu_{\mathbf{d}} + \alpha\sigma_{\mathbf{d}}$, where $\mu_{\mathbf{d}}$ represents the mean square distance of the data points and $\sigma_{\mathbf{d}}$ represents their standard deviation.

$$\Pr(|\overline{||\mathbf{d}_i||^2} - \mu_{\mathbf{d}}| \geq \alpha\sigma_{\mathbf{d}}) \leq \frac{1}{\alpha^2} \tag{7}$$

Eq. 7 indicates that the fraction of data points, whose square distance is greater than $\alpha$ times the standard deviation away from the mean, does not exceed $\frac{1}{\alpha^2}$. Therefore, this upper bound proxy offers a high coverage rate, which is further verified by real data as shown in Fig. 6. For simplicity, we approximate the bound by eliminating the dot product term because the expectation is 0, $\mathbb{E}[\langle\overline{\mathbf{d}_i}, \mathbf{n}\rangle] = 0$. The dataset upper bound is:

$$p_{ub} \approx \frac{1}{1 + (N-1)\exp\left(-\frac{||\mathbf{d}||^2}{2\sigma^2}\right)} \tag{8}$$

As shown in Fig. 6, as $\sigma$ increases, the decrease in $p_{ub}$ becomes progressively slower. Typically, the image dataset size $N$ is large, so we choose $\sigma_{max} = ||\mathbf{d}||$ to achieve the trade-off between the quality of generated data and NFE.

**NFE-guided sampling noise schedule.** The efficacy of a given numerical method is intimately tied to the chosen step size, which determines the resolution at which the ODE is approximated. Specifically, the local error of numerical methods is composed mostly from the truncation error and prediction error of the neural network $D_\theta(\mathbf{x}; \sigma)$.

Leveraging this insight, we investigate the relationship between the number of evaluation points (NEP) and the final performance. Tab. 4 shows that in the context of large step sizes, the second-order midpoint method often becomes preferable, given its capacity to provide a more representative perspective of the derivative $\frac{(\mathbf{x} - D_\theta(\mathbf{x};\sigma))}{\sigma}$ across the entire interval. Conversely, when operating with smaller step sizes, Heun's second-order method demonstrates superior performance. This can be attributed to its predictor-corrector approach, which effectively ensembles network predictions, mitigating the errors from the neural network $D_\theta(\mathbf{x}; \sigma)$. Notably, with an increasing NFE budget, opting for a third-order method, in lieu of further reducing the step size, emerges as a more optimal strategy to concurrently diminish truncation error and neural network error. The results demonstrate that the number of evaluation point is essential for fast ODE sampling of diffusion model.

Motivated by the observation, we propose a dynamic scheme for the hybrid numerical method and sampling noise schedule, $\sigma_{sample}(t)$, based on the NFE budget.

---

**Algorithm 1** Sampling process with dynamic numerical method and noise schedule guided by NFE.

1: **procedure** NFESAMPLER($D_\theta, NFE, \sigma_{max}, \sigma_{min}, h_\tau, \rho$)
2:  $\quad T \leftarrow \lfloor(NFE + 1)/2\rfloor, \lambda_{max} \leftarrow \log\sigma_{max}, \lambda_{min} \leftarrow \log\sigma_{min}$
3:  $\quad h \leftarrow (\lambda_{max} - \lambda_{min})/(T - 1)$ $\qquad\qquad$ ▷ Step size intended for 2nd-order method.
4:  $\quad$ **if** $h >= h_\tau$ **then**
5:  $\quad\quad isMidpoint \leftarrow$ True$, TOde3 \leftarrow 0$ $\qquad\qquad\qquad$ ▷ Midpoint method.
6:  $\quad$ **else**
7:  $\quad\quad T \leftarrow \lfloor(\lambda_{max} - \lambda_{min})/h_\tau + 1\rfloor$ $\qquad\qquad$ ▷ Step num intended for $h_\tau$.
8:  $\quad\quad TOde3 \leftarrow NFE - 2 * T + 1$ $\qquad$ ▷ 3rd-order step num in hybrid mode.
9:  $\quad\quad$ **if** $TOde3 > T - 1$ **then**
10: $\quad\quad\quad T \leftarrow \lfloor(NFE + 2)/3\rfloor$ $\qquad\qquad$ ▷ 3rd-order step num in pure mode.
11: $\quad\quad\quad TOde3 \leftarrow T - 1$
12: $\quad\quad$ **end if**
13: $\quad\quad isMidpoint \leftarrow$ False $\qquad$ ▷ Heun's method and 3rd-order method.
14: $\quad$ **end if**
15: $\quad \mathbf{x}_0 \leftarrow$ HYBRIDODESAMPLER($D_\theta, T, \sigma_{max}, \sigma_{min}, \rho, isMidpoint, TOde3$)
16: $\quad$ **return** $\mathbf{x}_0$

---

Here, $h_\tau$ represents the threshold for the step size. When the intended step size $h$ exceeds $h_\tau$ the 2nd-order midpoint method is employed. Conversely, the sampling noise schedule is adjusted for the hybrid approach utilizing both Heun's 2nd-order method and the 3rd-order method is adopted.

---

**Algorithm 2** Hybrid ODE Sampler

1: **procedure** HYBRIDODESAMPLER($D_\theta, T, \sigma_{max}, \sigma_{min}, \rho, isMidpoint, numOde3$)
2:  $\quad \sigma_{i>0} \leftarrow (\sigma_{min}^{\frac{1}{\rho}} + \frac{i-1}{N-1}(\sigma_{max}^{\frac{1}{\rho}} - \sigma_{min}^{\frac{1}{\rho}}))^\rho, \sigma_0 \leftarrow 0,$ **sample** $\mathbf{x}_N \sim \mathcal{N}(\mathbf{0}, \sigma_{max}^2\mathbf{I})$
3:  $\quad$ **for** $i \in \{T, ..., 1\}$ **do**
4:  $\quad\quad$ **if** $i = 1$ **then**
5:  $\quad\quad\quad \mathbf{x}_{i-1} \leftarrow$ ODESTEP($D_\theta, \mathbf{x}_i, \sigma_i, \sigma_{i-1}$, Euler) $\qquad$ ▷ The final step is Euler step.
6:  $\quad\quad$ **else if** $isMidpoint$ **then**
7:  $\quad\quad\quad \mathbf{x}_{i-1} \leftarrow$ ODESTEP($D_\theta, \mathbf{x}_i, \sigma_i, \sigma_{i-1}$, Midpoint) $\qquad$ ▷ Midpoint step for $h >= h_\tau$.
8:  $\quad\quad$ **else if** $i < N - numOde3 + 1$ **then**
9:  $\quad\quad\quad \mathbf{x}_{i-1} \leftarrow$ ODESTEP($D_\theta, \mathbf{x}_i, \sigma_i, \sigma_{i-1}$, Heun) $\qquad$ ▷ Heun's step in hybrid mode.
10: $\quad\quad$ **else**
11: $\quad\quad\quad \mathbf{x}_{i-1} \leftarrow$ ODESTEP($D_\theta, \mathbf{x}_i, \sigma_i, \sigma_{i-1}$, Ode3) $\qquad$ ▷ 3rd-order step: Large $\sigma$ first.
12: $\quad\quad$ **end if**
13: $\quad$ **end for**
14: $\quad$ **return** $\mathbf{x}_0$

---

In this context, $\rho$ controls the interpolation mode between $\sigma_{max}$ and $\sigma_{min}$, and the Euler method is invariably utilized in the final step. We prioritize allocating the NFE of the 3rd-order method to the steps where $\sigma$ is comparatively larger, as the first-order derivatives exhibit more rapid variations in those regions.

We assess the efficacy of the NFE-guided sampling noise schedule on the CIFAR-10 dataset. Given that our proposed sampling noise schedule necessitates no additional training, we opt to employ previously established, state-of-the-art pre-trained diffusion models (Karras et al., 2022). The initial noise distribution is determined by our Data-driven sampling noise schedule. As shown in Fig. 7, our Data-Driven NFE-Guided sampling noise schedule can greatly speed up the sampling of existing pre-trained diffusion models by adjusting step size and the order of ODE solver based on the NFE budget.

**Comparison of different ODEs.** One main source of errors introduced by the ODE numerical methods is discretization error, which is the difference between the true solution and the discrete approximation at each step. The step size of the mainstream sampling noise schedules is measured in $\log(\sigma)$ space, rather than $\sigma$ space, as shown in Fig. 1. The Reverse ODE in Eq. 1 can be reformulated as:

$$
\begin{aligned}
\lambda\text{-space:} \quad & \mathrm{d}\mathbf{x} = \epsilon_\theta(\mathbf{x}; \lambda)\exp(\lambda)\mathrm{d}\lambda \\
\sigma\text{-space:} \quad & \mathrm{d}\mathbf{x} = \epsilon_\theta(\mathbf{x}; \log \sigma)\mathrm{d}\sigma,
\end{aligned}
\tag{9}
$$

where $\lambda = \log \sigma$ and $\epsilon_\theta(\mathbf{x}; \log \sigma) = \frac{(\mathbf{x} - D_\theta(\mathbf{x};\sigma))}{\sigma}$, which is the $\epsilon$-prediction function in (Ho et al., 2020). In theory, the two ODEs are equivalent. In practice, the local truncation error varies in different spaces. Specifically, the Taylor expansion of the integration for a step is:

$$
\begin{aligned}
\lambda\text{-space: } \mathbf{x}_t - \mathbf{x}_s &= \sum_{k=0}^{n} \frac{h_\lambda^k}{k!} (\epsilon_\theta(\mathbf{x}_s; \lambda_s)\exp(\lambda_s))^{(k)} + \frac{h_\lambda^{n+1}}{(n+1)!}(\epsilon_\theta(\mathbf{x}_m; \lambda_m)\exp(\lambda_m))^{(n+1)} \\
\sigma\text{-space: } \mathbf{x}_t - \mathbf{x}_s &= \sum_{k=0}^{n} \frac{h_\sigma^k}{k!} \epsilon_\theta^{(k)}(\mathbf{x}_s; \log \sigma_s) + \frac{h_\sigma^{n+1}}{(n+1)!}\epsilon_\theta^{(n+1)}(\mathbf{x}_c; \log \sigma_c),
\end{aligned}
\tag{10}
$$

where $\cdot^{(k)}$ is the $k$-th derivative, $h_\lambda$ and $h_\sigma$ are the step size in $\lambda$-space and $\sigma$-space. The Lagrange remainder in the $\lambda$-space contains the $\exp(\lambda)$ term, which involves the $\epsilon_\theta(\mathbf{x})$ in high-order derivatives, making the truncation error remains even when the $\epsilon_\theta^{(n+1)}(\mathbf{x})$ is $\mathbf{0}$. As shown in Tab. 3, the results of 3rd-order $\lambda$-space ODE solver (Lu et al., 2022), DPM-Solver, are inferior to our $\sigma$-space 3rd-order ODE solver in the small step size scenario, i.e. the NFE is large.

## 6 CONCLUSION AND LIMITS

In conclusion, our explorations and findings have demonstrated substantial disparities in noise levels across images of different resolutions, significantly affecting the performance of the diffusion model. Motivated by these discrepancies, we develop the novel metric Weighted Signal-Noise-Ratio (WSNR) to proficiently quantify noise levels in the RGB space and successfully extend WSNR to the latent space. The proposed WSNR-Equivalent training noise schedule has been shown to improve the generative performance of diffusion models in both high-resolution RGB and latent spaces, setting a precedent in quantifying noise levels in the forward process of the diffusion model. Moreover, by leveraging an ODE solver under the probability flow ODE framework and conducting detailed analysis, we proposed a data-driven sampling noise schedule, which allows us to optimize the integration interval of the ODE, balancing efficiency and generation quality. Besides, we find the number of evaluation points to be crucial, necessitating dynamic selection strategies according to computational constraints to optimize generation quality, especially when dealing with diverse step sizes.

Limits: Our dynamic schedule only includes the explicit Runge-Kutta methods, such as Heun's method, Midpoint method and 3rd-order method. More advanced ODE solvers, such as DPM-Solver and DEIS Solver, are not involved in. We leave this for future work.

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

# A  APPENDIX

## A.1  THE DERIVATION OF EXPECTED POWER SPECTRUM OF GAUSSIAN NOISE

We use the definition of orthogonal power spectrum to proof the expected power spectrum of isotropic 2D Gaussian noise $\sigma\mathbf{n}$ is constant $\sigma^2$ at each frequency component.

$$
\begin{aligned}
\mathbb{E}[P_{\sigma\mathbf{n}}(u,v)] &= \sigma^2\mathbb{E}[F_{\mathbf{n}}(u,v)F_{\mathbf{n}}^*(u,v)] \\
&= \frac{\sigma^2}{HW}\mathbb{E}[(\sum_{k=0}^{H-1}\sum_{l=0}^{W-1}\mathbf{n}(k,l)e^{-j2\pi\left(\frac{ku}{H}+\frac{lv}{W}\right)})(\sum_{k'=0}^{H-1}\sum_{l'=0}^{W-1}\mathbf{n}(k',l')e^{j2\pi\left(\frac{k'u}{H}+\frac{l'v}{W}\right)})] \\
&= \frac{\sigma^2}{HW}\sum_{k=0}^{H-1}\sum_{l=0}^{W-1}\underbrace{\mathbb{E}[\mathbf{n}(k,l)^2]}_{=1} \\
&+ \frac{\sigma^2}{HW}\sum_{k=0}^{H-1}\sum_{l=0}^{W-1}\sum_{k'=0,k'\neq k}^{H-1}\sum_{l'=0,l'\neq l}^{W-1}\underbrace{\mathbb{E}[\mathbf{n}(k,l)\mathbf{n}(k',l')]}_{=0}e^{-j2\pi(\frac{ku}{H}+\frac{lv}{W})}e^{j2\pi(\frac{k'u}{H}+\frac{l'v}{W})} \\
&= \sigma^2,
\end{aligned}
\tag{11}
$$

where $F_{\mathbf{n}}$ is the 2D Discrete Fourier Transform.

## A.2  THE DERIVATION OF IDEAL SOLUTION AS WEIGHTED SUM OF CLEAN DATA POINT

Given the noisy data $\mathbf{x}$ with additional Gaussion noise $\sigma\mathbf{n}$, the probability of the noisy data $p(\mathbf{x};\sigma)$ can be expressed as:

$$
\begin{aligned}
p(\mathbf{x};\sigma) &= p_{\text{data}} * \mathcal{N}(0,\sigma^2\mathbf{I}) \\
&= \int_{\mathbb{R}^d} p_{\text{data}}(\mathbf{x}_0)\mathcal{N}(\mathbf{x};\mathbf{x}_0,\sigma^2\mathbf{I})dx_0 \\
&= \int_{\mathbb{R}^d}\left[\frac{1}{Y}\sum_{i=1}^{Y}\delta(\mathbf{x}_0-\mathbf{y}_i)\right]\mathcal{N}(\mathbf{x};\mathbf{x}_0,\sigma^2\mathbf{I})d\mathbf{x}_0 \\
&= \frac{1}{Y}\sum_{i=1}^{Y}\int_{\mathbb{R}^d}\mathcal{N}(\mathbf{x};\mathbf{x}_0,\sigma^2\mathbf{I})\delta(\mathbf{x}_0-\mathbf{y}_i)d\mathbf{x}_0 \\
&= \frac{1}{Y}\sum_{i=1}^{Y}\mathcal{N}(\mathbf{x};\mathbf{y}_i,\sigma^2\mathbf{I}),
\end{aligned}
\tag{12}
$$

where the clean data probability distribution $p_{\text{data}}(\mathbf{x}) = \frac{1}{Y}\sum_{i=1}^{Y}\delta(\mathbf{x}-\mathbf{y}_i)$ and the number of data points is $Y$. Since the denoiser $D(\mathbf{x};\sigma)$ aims to restore the clean image, the loss is expressed as:

$$
\begin{aligned}
\mathcal{L}(D;\sigma) &= \mathbb{E}_{\mathbf{y}\sim p_{\text{data}}}\mathbb{E}_{\mathbf{n}\sim\mathcal{N}(0,\sigma^2\mathbf{I})}\|D(\mathbf{y}+\mathbf{n};\sigma)-\mathbf{y}\|_2^2 \\
&= \mathbb{E}_{\mathbf{y}\sim p_{\text{data}}}\mathbb{E}_{\mathbf{x}\sim\mathcal{N}(\mathbf{y},\sigma^2\mathbf{I})}\|D(x;\sigma)-y\|_2^2 \\
&= \mathbb{E}_{\mathbf{y}\sim p_{\text{data}}}\int_{\mathbb{R}^d}\mathcal{N}(\mathbf{x};\mathbf{y},\sigma^2\mathbf{I})\|D(\mathbf{x};\sigma)-\mathbf{y}\|_2^2\,d\mathbf{x} \\
&= \frac{1}{Y}\sum_{i=1}^{Y}\int_{\mathbb{R}^d}\mathcal{N}(\mathbf{x};\mathbf{y}_i,\sigma^2\mathbf{I})\|D(\mathbf{x};\sigma)-\mathbf{y}_i\|_2^2\,d\mathbf{x} \\
&= \int_{\mathbb{R}^d}\frac{1}{Y}\sum_{i=1}^{Y}\mathcal{N}(\mathbf{x};\mathbf{y}_i,\sigma^2\mathbf{I})\|D(\mathbf{x};\sigma)-\mathbf{y}_i\|_2^2\,d\mathbf{x}
\end{aligned}
\tag{13}
$$

The solution is obtained by setting the gradient w.r.t $D(\mathbf{x}; \sigma)$ to zero:

$$0 = \nabla_{D(\mathbf{x};\sigma)} \mathcal{L}(D; \mathbf{x}, \sigma)$$

$$0 = \nabla_{D(\mathbf{x};\sigma)} \left[ \frac{1}{Y} \sum_{i=1}^{Y} \mathcal{N}(\mathbf{x}; \mathbf{y}_i, \sigma^2 \mathbf{I}) \| D(\mathbf{x}; \sigma) - \mathbf{y}_i \|_2^2 \right]$$

$$0 = \sum_{i=1}^{Y} \mathcal{N}(\mathbf{x}; \mathbf{y}_i, \sigma^2 \mathbf{I}) \nabla_{D(\mathbf{x};\sigma)} \left[ \| D(\mathbf{x}; \sigma) - \mathbf{y}_i \|_2^2 \right]$$

$$0 = \sum_{i=1}^{Y} \mathcal{N}(\mathbf{x}; \mathbf{y}_i, \sigma^2 \mathbf{I}) \left[ 2D(\mathbf{x}; \sigma) - 2\mathbf{y}_i \right]$$

$$0 = \left[ \sum_{i=1}^{Y} \mathcal{N}(\mathbf{x}; \mathbf{y}_i, \sigma^2 \mathbf{I}) D(\mathbf{x}; \sigma) \right] - \left[ \sum_{i=1}^{Y} \mathcal{N}(\mathbf{x}; \mathbf{y}_i, \sigma^2 \mathbf{I}) \mathbf{y}_i \right] \quad (14)$$

$$D(\mathbf{x}; \sigma) = \frac{\sum_{i=1}^{Y} \mathcal{N}(\mathbf{x}; \mathbf{y}_i, \sigma^2 \mathbf{I}) \mathbf{y}_i}{\sum_{i=1}^{Y} \mathcal{N}(\mathbf{x}; \mathbf{y}_i, \sigma^2 \mathbf{I})}$$

$$D(\mathbf{x}; \sigma) = \frac{\sum_{i=1}^{Y} \exp(\frac{\|\mathbf{x} - \mathbf{y}_i\|^2}{-2\sigma^2}) \mathbf{y}_i}{\sum_{i=1}^{Y} \exp(\frac{\|\mathbf{x} - \mathbf{y}_i\|^2}{-2\sigma^2})}$$

$$D(\mathbf{x}; \sigma) = \sum_{i=1}^{Y} \text{softmax}(\frac{\|\mathbf{x} - \mathbf{y}_i\|^2}{-2\sigma^2}) \mathbf{y}_i$$

As shown in Eq. 14, the ideal solution for the denoiser $D(\mathbf{x}; \sigma)$ can be found as the weighted sum of all clean data in the dataset.

### A.3 THE DETAILED IMPLEMENTATION OF OUR ODE STEP

To provide a clear depiction of our applied ODE method, we present the pseudo code in Algo 3 and the Butcher tableau in Tab. 5. The Butcher tableau is instrumental in detailing the numerical integrators used for solving the ODEs and offers a concise representation of the Runge-Kutta methods applied in our experiments. In relation to step size determination, we introduce a nuanced approach, where the step size exists in the $\log(\sigma)$ space. The transition to $\log(\sigma)$ space ensures that efficiency of sampling.

---

**Algorithm 3** ODE Step

---

1: **procedure** ODESTEP($D_\theta, \mathbf{x}_i, \sigma_i, \sigma_{i-1}, algo$)
2:     $\mathbf{c}, \mathbf{b}, \mathbf{A} \leftarrow butcherTable[algo]$
3:     $kList \leftarrow [], nOrder \leftarrow len(\mathbf{c})$
4:     **for** $u = 1$ **to** $nOrder$ **do**
5:         $\sigma \leftarrow \exp(\lambda_i + \mathbf{c}_u(\lambda_{i-1} - \lambda_i)), \mathbf{k} \leftarrow \mathbf{0}$
6:         **for** $v = 1$ **to** $u - 1$ **do**
7:             $\mathbf{k} \leftarrow \mathbf{k} + kList[v] * \frac{\mathbf{A}_{u,v}}{\mathbf{c}_u}$
8:         **end for**
9:         $\mathbf{x} \leftarrow \mathbf{x}_i + \mathbf{k}(\sigma - \sigma_i)$
10:        Append $\frac{\mathbf{x} - D_\theta(\mathbf{x}, \sigma)}{\sigma}$ to $kList$
11:     **end for**
12:     $\mathbf{x}_{i-1} \leftarrow \mathbf{x}_i + (\sigma_{i-1} - \sigma_i) \sum_{u=1}^{nOrder} kList[u] * \mathbf{b}_u$
13:     **return** $x_{i-1}$

---

Table 5: Butcher Tableau for ODE Numerical Methods

| **Euler** | **Midpoint** | **Heun** | **Ode3** |
|---|---|---|---|

$$
\begin{array}{c|c}
0 & \\
\hline
 & 1
\end{array}
\qquad
\begin{array}{c|cc}
0 & & \\
1/2 & 1/2 & \\
\hline
 & 0 & 1
\end{array}
\qquad
\begin{array}{c|cc}
0 & & \\
1 & 1 & \\
\hline
 & 1/2 & 1/2
\end{array}
\qquad
\begin{array}{c|ccc}
0 & & & \\
1 & 1 & & \\
1/2 & 1/4 & 1/4 & \\
\hline
 & 1/6 & 1/6 & 2/3
\end{array}
$$

## A.4 Is Equation 1 compatible with Variance Preserving?

Our ODE (Eq. 1) is compatible with Variance Preserving (VP) case. As discussed in Eq.7 of (Karras et al., 2022), we can write the denoiser $D(\mathbf{x}; \sigma)$ in the following form:

$$
D_\theta(\mathbf{x}; \sigma) = c_{\text{skip}}(\sigma)x + c_{\text{out}}(\sigma)F_\theta(c_{\text{in}}(\sigma)\mathbf{x}; c_{\text{noise}}(\sigma)) \tag{15}
$$

For VP case, the corresponding config is $c_{\text{skip}}(\sigma) = 1$, $c_{\text{out}}(\sigma) = -\sigma$, $c_{\text{in}}(\sigma) = \frac{1}{\sqrt{\sigma^2+1}}$, $c_{\text{noise}}(\sigma) = (M-1)\sigma^{-1}(\sigma)$. This implies that our Eq.1 is compatible with the VP case and our sampling noise schedule can be evaluated with the pre-trained VP model. As for the training process, we follow the training loss in (Karras et al., 2022), setting the training target as:

$$
\mathcal{L} = \mathbb{E}_{\sigma, \mathbf{y}, \mathbf{n}} \left[ \overbrace{\lambda(\sigma)c_{\text{out}}(\sigma)^2}^{\text{effective weight}} \left\| \overbrace{F_\theta\left(c_{\text{in}}(\sigma) \cdot (\mathbf{y} + \mathbf{n}); c_{\text{noise}}(\sigma)\right)}^{\text{network output}} - \overbrace{\frac{1}{c_{\text{out}}(\sigma)}(\mathbf{y} - c_{\text{skip}}(\sigma) \cdot (\mathbf{y} + \mathbf{n}))}^{\text{effective training target}} \right\|^2 \right] \tag{16}
$$

where the $\lambda(\sigma)$ is the loss weight for each noise level. Therefore, we believe that our training process is compatible with interpreting the VP case.

