# OpenReview forum: "Rethinking the Noise Schedule of Diffusion-Based Generative Models"
_ICLR.cc/2024/Conference — Submitted to ICLR 2024_

### Official Review · Reviewer_nhAz · 2023-10-30

**Soundness:** 3 good
**Presentation:** 2 fair
**Contribution:** 3 good
**Rating:** 6
**Confidence:** 3

**Summary:**

In this paper, the authors investigate the training/sampling noise schedule through the lens of power spectrum and introduce the weighted signal-noise ratio (WSNR). The authors show that adjusting the noise schedule according to WSNR is able to improve the performance of high-resolution image generation and ODE-based sampling.

**Strengths:**

1) Although the importance of noise schedule has been studied in previous papers, most of previous methods adjusting the schedule intuitively. The authors proposed a numerical metric and validated the effectiveness of the proposed metric.
2) The motivation of this paper is clear and the organization and presentation of this paper is good.
3) The experimental results validated the advantage of adjusting noise schedule for high-resolution image generation and ODE-based sampling.

**Weaknesses:**

1) The authors proposed the WSNR metric and adjusting the noise schedule of high-resolution image generation to align the WSNR schedule with low-resolution image, although the authors show that such adjustment is beneficial for high-resolution image generation, a more important question is whether the proposed metric could shed light on optimal schedule for image generation. Since the schedule for 64\times 64 image generation is also intuitively setted, why should we align the schedule of high-resolution image generation to 64 \times 64?
2) The idea of data-driven ODE noise schedule is interesting, and the authors show that the proposed method is able to improve the sampling quality. Is the newly proposed sampling strategy highly related to the WSNR metric, can we adjust the sampling schedule based on other metrics such as PSNR? The improvement is mainly due to the data-driven framework or the newly proposed metric.

**Questions:**

Please refer to the weakness part.

---

> ### Author Response · Authors · 2023-11-21
>
> We thank the reviewer for the positive and comprehensive feedback. We are encouraged that the reviewer found our WSNR noise schedule is beyond adjusting noise schedule intuitively. In the following, we provide a response to the questions in the review.
>
>
> Q1: Why should we align the schedule of high-resolution image generation to 64 x 64? Whether the proposed metric could shed light on optimal schedule for image generation?
>
> A1:  We appreciate the reviewer’s query regarding our anchor noise schedule choice.
>
> We chose the 64x64 noise schedule as our anchor training noise schedule because the diffusion model is well-trained at this resolution. However, the choice of resolution for alignment is flexible.
>
> Our WSNR metric lays the foundation for finding the optimal schedule for image generation, reducing our exploration to just one set of hyperparameters. This is because the WSNR-Equivalent Training Noise Schedule consistently achieves superior results with a single set of hyperparameters. In contrast, other noise schedules fail to consistently achieve desired performance across multiple resolutions in both RGB and latent spaces under a single hyperparameter set.
>
> Furthermore, we believe that our proposed data-driven sampling schedule can aid in exploring the design of training noise schedules. Since probabilistic generative models are essentially fitting data distributions, we hold that the optimal noise schedule should be data-driven. We leave this exploration to our future work.
>
> Q2: Is the newly proposed sampling strategy highly related to the WSNR metric? Can we adjust the sampling schedule based on other metrics such as PSNR?
>
> A2: We are very grateful to the reviewer for raising such insightful questions, which have sparked our thinking about how to use WSNR to uniformly design both training and sampling noise schedules.
>
> Our proposed WSNR metric can be used to uniformly represent noise levels across different scales, meaning that a data-driven sampling schedule can be designed and expressed through WSNR. We found that for the FFHQ-64 and FFHQ-256 datasets, the ODE integration intervals should be [0.002, 98.96] and [0.008, 394.62], respectively. When represented in terms of WSNR, these intervals become [117166875, 0.0478] and [117166875, 0.0481], showing that FFHQ 64 and 256 are similar in terms of the WSNR metric.
>
> We have not explored replacing WSNR with PSNR, because WSNR is a promising metric. We will leave the exploration regarding PSNR for future work.
>
> ------------------------------------------------------------------------
>
> We would greatly appreciate it if you could confirm whether the revisions made align with your expectations and satisfactorily resolve the issues previously highlighted.
>
> Your further guidance and feedback will be important to us and we are committed to making any additional improvements as needed.
>
> Thank you very much for your time and consideration. We look forward to your response.

---

### Official Review · Reviewer_pugY · 2023-11-04

**Soundness:** 2 fair
**Presentation:** 3 good
**Contribution:** 3 good
**Rating:** 5
**Confidence:** 4

**Summary:**

This paper investigates noise scheduling strategies within the scope of denoising diffusion generative models. They investigate the training noise schedule through the lens of power spectrum and introduce a novel metric, weighted signal-noise-ratio, to uniformly represent the noise level in both RGB and latent spaces, enhancing the performance of high-resolution models in these spaces with WSNR-Equivalent training noise schedules.
They explore the correlation between the number of evaluation points and the generation quality to optimize the acceleration of the ODE solver in the diffusion model. Based on practical considerations of evaluation point effects, we propose an adaptive scheme to choose numerical methods within computational constraints, balancing efficacy and efficiency.

**Strengths:**

1. This paper views the noise scheduling problem from the perspective of power spectra of various frequency components and discover that the average power spectra of isotropic Gaussian noise are consistent across all components.

2. The proposed metric, WSNR, quantifies the noise level of the training data in both the RGB space and latent space.

3. It empirically explores the relationship between the number of evaluation points and the generation quality.

**Weaknesses:**

1. The noise scheduling is discussed in previous works from different perspectives. The concurrent work [1] also discusses the noise schedule from the spectra view. The authors are encouraged to discuss the differences.
[1] Relay diffusion: Unifying diffusion process across resolutions for image synthesis

2. This paper aims to solve the terrible performance of existing noise scheduling in high resolutions. But the experiments are all conducted on small resolutions, with the highest resolutions being 256x256.  Experiments with a higher resolution are highly recommended.

**Questions:**

Please refer to the weaknesses and questions.

---

> ### Author Response · Authors · 2023-11-21
>
> Q1: The concurrent work [1] also discusses the noise schedule from the spectra view. The authors are encouraged to discuss the differences.
>
> A1:   Thank you for bringing concurrent work [1] to our attention. We acknowledge that [1] also explores the noise level from a spectral perspective in RGB space.
>
> **However, our work and [1] have several distinct differences**:
> 1. **Different Approaches**: Our work is dedicated to designing a robust training noise schedule applicable across different resolutions and latent spaces. [1] mainly focuses on the relay connection within their proposed RDM, enabling a cascading sampling process from 64x64 to 256x256 in a single pass. It seems that they do not contribute to the training noise schedule. If my understanding of [1] is incorrect, please kindly correct me.
> 2. **Different Analysis Metrics**: We propose WSNR as a consistent quantization metric, ensuring consistent quantization results at the same noise level across different resolutions. In contrast, [1] uses the SNR quantization metric, which is not consistent in different resolutions.
> 3. **Different Analytical Perspectives**: We analyze the average power of images and Gaussian noise from a power spectrum perspective, highlighting that the average power of Gaussian noise is uniform across all frequency components. This conclusion is not stated in their work.
>
> Additionally, it's worth noting that we have conducted theoretical analysis on the sampling noise schedule and proposed improvements.
> Moreover, [1] is also a submission to ICRL 2024. We believe these are two independent pieces of research work, and this should not be considered a weakness or even a reason for rejection.
>
>
>
> Q2: The noise schedule is discussed in previous works from different perspectives.
>
> A2:  We appreciate the reviewer's comment regarding the discussion of noise scheduling in prior works.
> Since our work is titled "Rethinking the Noise Schedule of Diffusion-Based Generative Models", we have acknowledged that some previous works have explored noise schedule design. However, our paper has **unique contributions**:
> - We propose a novel WSNR metric to consistently quantify the noise level across multiple resolutions in **RGB space and latent space** for the forward process of diffusion models.
> - We significantly improve the performance of diffusion models, via our novel WSNR-Equivalent training noise scheduler. Specifically, our method decreases the FID score by 3.6 on FFHQ-256x256 dataset, and achieves the same FID scores on ImageNet-256 and 512 for UViT-M (#Params: 287M)  as those of UViT-L (#Params: 287M).
> - We analyze that the **ODE sampling noise schedule** should be data-driven. Consequently, we propose estimating the integration interval based on the average data distance.
> - We explore the relationship between the number of evaluation points and generation quality. Our findings lead us to develop a **NFE-guided sampling noise schedule**.
> - Our sampling schedule refines the FID of pre-trained CIFAR-10 and FFHQ-64 models from 1.92 and 2.45 to 1.89 and 2.25, respectively, utilizing 35 network evaluations per image.
>
>
> As the reviewer did not provide further explanation and references about the "previous works" and "different perspectives", we will discuss the recent representative research works [2,3,4,5,6], which proposed handcrafted training noise schedules to improve the performance of diffusion models. However, the proposed schedules are based on the SNR metric, which is neither consistent in different resolutions, nor consistent in latent space.
>
> That means their schedules **fail to** achieve desired performance consistently on various across multiple resolutions in RGB space and latent space under a single set of hyperparameters. In contrast, **ours can achieve superior results** consistently with just one set of hyperparameters.
>
> Besides, our analysis and improvement on ODE sampling noise schedule, an aspect not addressed by the other methods. This unique focus not only demonstrates our method's innovative approach but also highlights its ability to fill a critical gap in the current research landscape. By delving into areas previously unexplored, our work contributes significantly more to the advancement of the field, offering novel insights and practical improvements that set a new standard for future research in this area.

---

> ### Author Response · Authors · 2023-11-21
>
> Q3: The highest resolution is 256x256. Experiments with a higher resolution are highly recommended.
>
>
> A3: We thank the reviewer for highlighting the importance of conducting experiments at higher resolutions, especially given our paper's focus on improving noise scheduling in high-resolution contexts.
>
> Due to the limited time for rebuttal and our computational resources, we trained a network on ImageNet 512x512 to demonstrate the effectiveness of our training schedule. Specifically, we use UViT-M as network architecture, similar to the experiment in our draft on ImageNet 256x256, keeping the hyperparameters unchanged. The results, as shown in the table below, indicate that UViT-M, utilizing our WSNR-Equivalent noise schedule, achieves a superior FID score compared to the UViT-L model, which has more than double the parameter count of UViT-M.
>
> | Network | #Param | FID |
> | --- | --- | --- |
> | UViT-M, p(WSNR) | **131M** | **4.55** |
> | UViT-L, p($\sigma$) | 287M | 4.67 |
>
>
>
> We understand the importance of ensuring that our paper meets the high standards of ICLR and would greatly appreciate it if you could confirm whether the revisions made align with your expectations and satisfactorily resolve the issues previously highlighted.
>
> Your further guidance and feedback will be important to us and we are committed to making any additional improvements as needed.
>
> Thank you very much for your time and consideration. We look forward to your response.
>
>
>
> [1] Teng, Jiayan, et al. "Relay diffusion: Unifying diffusion process across resolutions for image synthesis." arXiv preprint arXiv:2309.03350 (2023).
>
> [2] Nichol, Alexander Quinn, and Prafulla Dhariwal. "Improved denoising diffusion probabilistic models." International Conference on Machine Learning. PMLR, 2021.
>
> [3] Kingma, Diederik, et al. "Variational diffusion models." Advances in neural information processing systems 34 (2021): 21696-21707.
>
> [4] Choi, Jooyoung, et al. "Perception prioritized training of diffusion models." Proceedings of the IEEE/CVF Conference on Computer Vision and Pattern Recognition. 2022.
>
> [5] Chen, Ting. "On the importance of noise scheduling for diffusion models." arXiv preprint arXiv:2301.10972 (2023).
>
> [6] Hang, Tiankai, et al. "Efficient diffusion training via min-snr weighting strategy." arXiv preprint arXiv:2303.09556 (2023).

---

> ### Author Response · Authors · 2023-11-22
>
> Dear Reviewer pugY,
>
> We appreciate the diligent efforts put into reviewing our work.
>
> We have compared our novel training and sampling noise schedule with both previous and concurrent works. Besides, we conducted experiments on a higher resolution.  Furthermore, we improved the writing quality in our revised manuscript.
>
> We would greatly appreciate it if you could confirm whether the revisions made align with your expectations and satisfactorily resolve the issues previously highlighted.
>
> We sincerely look forward to your response and would happily address any other questions.
>
> Thank you,
>
> The authors

---

> ### Author Response · Authors · 2023-11-23
>
> Dear Reviewer pugY,
>
> Just a reminder, there are only 4 hours left for the rebuttal.
>
> After this time, we will not be able to participate in the discussion any further.
>
> We are waiting for your response. We would happily address any other questions.
>
> Best Regards,
>
> The Authors

---

> ### Author Response · Authors · 2023-11-23
>
> Dear Reviewer pugY,
>
> Just a reminder, there are only **3 hours** left for the rebuttal.
>
> After this time, we will not be able to participate in the discussion any further.
>
> We are waiting for your response. We would happily address any other questions.
>
> Best Regards,
>
> The Authors

---

> ### Author Response · Authors · 2023-11-23
>
> Dear Reviewer,
>
> Just a reminder, there are only **2 hours** left for the rebuttal.
>
> After this time, we will not be able to participate in the discussion any further.
>
> We are waiting for your response. We would happily address any other questions.
>
> Best Regards,
>
> The Authors

---

### Official Review · Reviewer_9Ms4 · 2023-11-04

**Soundness:** 3 good
**Presentation:** 2 fair
**Contribution:** 3 good
**Rating:** 5
**Confidence:** 3

**Summary:**

This paper provides a theoretical and empirical analysis of the noise schedule strategy in denoising diffusion generative models. The authors investigate training noise schedules from the perspective of power spectra and introduce a new metric called Weighted Signal-to-Noise Ratio (WSNR) to uniformly represent noise levels in both RGB space and latent space, improving the performance of high-resolution models. They also explore the inverse sampling process using the framework of Ordinary Differential Equations (ODEs), revealing the concept of optimal denoisers and providing insights into data-driven sampling noise schedules. Additionally, they explore the correlation between the number of evaluation points and the quality of generated samples, and propose optimizations for accelerating ODE solvers. The proposed method improves the FID of CIFAR-10 and FFHQ-64 models without requiring additional training.

**Strengths:**

- The authors propose a novel metric, weighted signal-noise-ratio (WSNR), to quantify the noise level in both RGB and latent spaces.
  - WSNR is an intuitive metric. Figure 2 helps to understand the motivation.
- They explore the correlation between the number of evaluation points and the generation quality, and propose a strategy to dynamically select numerical methods for better generation quality.
- They achieve improved performance in high-resolution RGB and latent spaces without additional training.
- They contribute to the field by quantifying the noise level of the forward process of the diffusion model and extending it to the latent space.
- They present empirical results on CIFAR-10, FFHQ-64, ImageNet-64, and FFHQ-256 datasets, demonstrating the effectiveness of the proposed methods.
- They discuss the probability of the synthesized data and the importance of a broad variety in generated samples.
- They introduce a data-driven sampling noise schedule to ensure the diversity of generated data.
- They identify the trade-off between the quality of generated data and the number of neural function evaluations (NFE) and proposes an appropriate value for the integration range.

**Weaknesses:**

- The motivation or justification for rethinking the noise schedule of diffusion-based generative models is not clearly explained in the introduction.
  - I could not understand how Figure 1 is related to the main motivation of the paper. Figure 2 was more intuitive.
- From Eq. (1), it seems to implicitly assume the variance exploding (VE) case, but it is not clear what happens in the variance preserving (VP) case.
- Overall, writing should be improved. In the current form, motivation is not clearly explained in the introduction, and it is not until Figure 2 in Section 4 that the motivation is understood.

**Questions:**

As I described in the weakness section, Eq. (1) seems to implicitly assume the VE case, but how abound the VP case?

---

> ### Author Response · Authors · 2023-11-21
>
> We are grateful to the reviewer for the comprehensive feedback. We are encouraged by the recognition that our WSNR metric is both novel and well-motivated, and that the proposed sampling schedule is efficient and effective. Below, we respond to the questions raised in the review.
>
>
> Q1: The motivation or justification for rethinking the noise schedule of diffusion-based generative models is not clearly explained in the introduction. How Figure 1 is related to the main motivation of the paper?
>
> A1: We appreciate the reviewer’s feedback regarding the clarity of our paper’s motivation and the role of Fig 1. We acknowledge an error in our initial submission regarding the referencing of figures. This analysis is actually presented in Fig 2. We apologize for any confusion this may have caused and have corrected this reference in our revised manuscript. Fig 1 is intended to illustrate the training noise schedule and sampling noise schedule.
>
> We have revised the introduction section to explain the motivation clearly.
> Here, we summarize our motivation briefly:
>
> 1. As shown in Fig. 2, significant variation in noise levels across images of different resolutions when exposed to Gaussian noise with identical standard deviations, leads to a discrepancy in the noise levels focused on by manually designed training noise schedules in different RGB space resolutions and latent space. Motivated by this observation, we propose a novel metric WSNR to quantify the noise level consistently across multiple resolutions in RGB space and latent space. We further propose a novel training noise schedule based on the WSNR metric.
>
> 2. As illustrated in Fig. 4 and elaborately explained in Sec. 5, the diversity of the generated data is jointly influenced by the initial Gaussian distribution at the start point and the Euclidean distance from the data points in the dataset. Therefore, we propose a data-driven sampling noise schedule to determine the integration interval of the diffusion ODE.
>
> 3. As demonstrated in Tab. 4, an increased number of evaluation points (the unique time steps at which model makes predictions) leads to better results when the step size is relatively large. Conversely, with smaller step sizes, the advantage of adding more evaluation points becomes less significant. Drawing on these findings, we propose a strategy for dynamically selecting numerical methods according to computational constraints, aiming to optimize generation quality.
>
>
>
> Q2: From Eq. (1), it seems to implicitly assume the variance exploding (VE) case, but it is not clear what happens in the variance preserving (VP) case.
>
> A2: Our proposed methods are applicable to VP case. As shown in Tab.3, our data-driven sampling noise schedule works when it is applied to DPM-Solver, which is based on VP ODE.
>
> Besides, we conduct experiments on VP cases to evaluate our WSNR-Equivalent training noise schedule. The FID score on FFHQ dataset in the following table show that our method further improves performance on VP cases.
>
> | FFHQ | 64 | 128 | 256 |
> | --- | --- | --- | --- |
> | $p(\text{WSNR})$ | 3.80 | 6.23 | 7.99 |
> | $p(\sigma)$ | 3.80 | 7.41 | 11.78 |
>
>
> Actually, the denoiser D_theta (x) in Eq. 1 can be used to explain VP, VE and EDM cases.
> As shown in Eq. 7 in [1], we can write the $D_\theta(x; \sigma)$ in the following form:
>
> $$D_\theta(x; \sigma) = c_{\text{skip}}(\sigma) x + c_{\text{out}}(\sigma) F_\theta(c_{\text{in}}(\sigma) x; c_{\text{noise}}(\sigma))$$
> For VP, $c_{\text{skip}}(\sigma)=1, c_{\text{out}}(\sigma)=-\sigma, c_{\text{in}}(\sigma)=\frac{1}{\sqrt{\sigma^2 + 1}}, c_{\text{noise}}(\sigma) = (M-1)\sigma^{-1}(\sigma)$.
> This implies that our Eq.1 is compatible with the VP case and our sampling noise schedule can be evaluated with  the pre-trained VP model.
>
>
> As for the training process, we follow the training loss in [1], setting the training target as:
>
> $E_{\sigma, y, n} \left[ \overbrace{\lambda(\sigma) c_{\text{out}}(\sigma)^2}^{\text{effective weight}} \left\| \overbrace{F_\theta \left( c_{\text{in}}(\sigma) \cdot (y + n); c_{\text{noise}}(\sigma) \right)}^{\text{network output}} - \overbrace{\frac{1}{c_{\text{out}}(\sigma)} (y - c_{\text{skip}}(\sigma) \cdot (y + n))}^{\text{effective training target}} \right\|^2 \right]$
>
> Therefore, we believe that our training process is compatible with interpreting the VP case.
>
> [1] Karras, Tero, et al. "Elucidating the design space of diffusion-based generative models." Advances in Neural Information Processing Systems 35 (2022).
>
>
> We understand the importance of ensuring that our paper meets the high standards of ICLR and would greatly appreciate it if you could confirm whether the revisions made align with your expectations and satisfactorily resolve the issues previously highlighted.
>
> Your further guidance and feedback will be important to us and we are committed to making any additional improvements as needed.
>
> Thank you very much for your time and consideration. We look forward to your response.

---

> ### Author Response · Authors · 2023-11-22
>
> Dear Reviewer 9Ms4,
>
> We appreciate the diligent efforts put into reviewing our work.
>
> We have tried to answer all your question about the Eq.1 and would greatly appreciate it if you could confirm whether the revisions made align with your expectations and satisfactorily resolve the issues previously highlighted.
>
> As you recognize so many strengths of our work, we sincerely look forward to your response.
>
> We would happily address any other questions.
>
> Thank you,
>
> The authors

---

> ### Author Response · Authors · 2023-11-23
>
> Dear Reviewer 9Ms4,
>
> Just a reminder, there are only 4 hours left for the rebuttal.
>
> After this time, we will not be able to participate in the discussion any further.
>
> We are waiting for your response. We would happily address any other questions.
>
> Best Regards,
>
> The Authors

---

> ### Author Response · Authors · 2023-11-23
>
> Dear Reviewer 9Ms4,
>
> Just a reminder, there are only **3 hours** left for the rebuttal.
>
> After this time, we will not be able to participate in the discussion any further.
>
> We are waiting for your response. We would happily address any other questions.
>
> Best Regards,
>
> The Authors

---

> ### Author Response · Authors · 2023-11-23
>
> Dear Reviewer,
>
> Just a reminder, there are only **2 hours** left for the rebuttal.
>
> After this time, we will not be able to participate in the discussion any further.
>
> We are waiting for your response. We would happily address any other questions.
>
> Best Regards,
>
> The Authors

---

### Official Review · Reviewer_jyko · 2023-11-04

**Soundness:** 4 excellent
**Presentation:** 4 excellent
**Contribution:** 3 good
**Rating:** 8
**Confidence:** 4

**Summary:**

This research study identifies substantial disparities  in noise levels across images of different resolutions,
significantly affecting the performance of the diffusion model. The manuscript then investigates the training of
diffusion models using a weighted signal-to-noise-ratio (WSNR) metric. This metric does not depend on the image
resolution. WSNR is shown to be a better metric to quantize the noise level in the forward diffusion process.
The manuscript provides the analysis of the diffusion model from the point of view of the ordinary differential equations
probability flows in Section 5

**Strengths:**

- The manuscript propose a weighted signal-to-noise-ration (WSNR) metric for training diffusion models which does not depend on the image  resolution.
- WSNR is shown to be a better metric to quantize the noise level in the forward diffusion process.
- Experimental results show that WSNR represents a valid metric to illustrate noise levels in the latent space.

**Weaknesses:**

-

**Questions:**

- How would the metric depend on the local properties of the image, such as the presence of flat regions or textures?
For example in Figure 2 the noise is evident in the background but it is masked in the region of the main object which is highly textured.

---

> ### Author Response · Authors · 2023-11-21
>
> We are thankful to the reviewer for their very positive and comprehensive feedback. We feel encouraged by their acknowledgment that our WSNR metric effectively quantifies the noise level, independent of image resolution, and is also applicable to latent space.
>
> Q1: How would the metric depend on the local properties of the image, such as the presence of flat regions or textures?
>
> A1: We are very grateful to the reviewer for raising such insightful questions, which is an important property for both neural networks and image processing.
>
> The WSNR metric, designed based on power spectrum analysis, quantifies noise levels at the image level. The frequency characteristics of local areas influence the overall power spectrum, thereby affecting the WSNR.
>
> Local properties are a common and crucial property for neural networks. We can further use Short-Time Fourier Transform (STFT) to analyze the power spectrum of local areas more directly and likewise use WSNR to represent the noise level in these areas. We plan to include this analysis in the final version of our work.
>
>
> Thank you very much for your time, consideration and support.

---

### Official Review · Reviewer_QVZ8 · 2023-11-06

**Soundness:** 3 good
**Presentation:** 3 good
**Contribution:** 3 good
**Rating:** 6
**Confidence:** 3

**Summary:**

This paper investigates the training noise schedule of diffusion models from the perspective of the spectrum. It introduces the weighted signal-noise ratio (WSNR) to better represent the noise level of latent variables of diffusion models. This paper also proposes an adaptive sampling scheme that better balances efficacy and efficiency.

**Strengths:**

1. The proposed WSNR can better measure the noise level of diffusion latent variables across various resolutions. Models trained with a WSNR-oriented schedule can generalize better to more resolutions.

2. The proposed adaptive sampling strategy better balances the efficacy and efficiency of diffusion models. It improves the performance of diffusion models without additional training.

**Weaknesses:**

1. The proposed WSNR-Equivalent training noise schedule and data-driven sampling noise schedule seem to be independent of each other， which weakns the focus of this paper.

2. Experiments in Table 1 and Table 2 compare with only EDM training noise schedule. The authors are suggested to compare with more training noise schedules to further verify the effectiveness of training noise schedule.

**Questions:**

See the weaknesses above.

---

> ### Author Response · Authors · 2023-11-21
>
> We thank the reviewer for the positive and comprehensive feedback. We are encouraged that the reviewer found our WSNR metric represents noise level well and our adaptive sampling scheme balances efficacy and efficiency well. In the following, we provide a response to the questions in the review.
>
>
> Q1: The proposed WSNR-Equivalent training noise schedule and data-driven sampling noise schedule seem to be independent of each other, which weakens the focus of this paper.
>
> A1: We respectively disagree with this viewpoint.
> Firstly, the training and sampling processes of diffusion models are strongly interrelated. Diffusion models are trained independently for each noise level. The inference stage can be seen as refining the generated images multiple times at various timesteps using the network. We believe that optimizing both the training and sampling stages simultaneously **highlights our contribution**.
>
> Secondly, exploring the training and sampling processes of diffusion models within a single work is **not a weakness**. In the **NIPS 2022 Outstanding Paper** [1], the authors **similarly improved the training and sampling processes of diffusion models**, which was not considered a weakness by the reviewers, area chairs, or senior area chairs.
>
> Finally, combining our methods will **further enhance the performance**.  We evaluate our trained model on FFHQ-256 using our proposed sampling method. The table shows that our sampling method consistently lowers the FID score of the trained model, indicating improved performance since a lower FID score is desirable.
>
> |  | Heun (NFE=35) | Ours (NFE=35) |
> | --- | --- | --- |
> | p(WSNR) | 7.89 | **7.40** |
> | p($\sigma$) | 11.49 | **11.02** |
>
>
> [1] Karras, Tero, et al. "Elucidating the design space of diffusion-based generative models." Advances in Neural Information Processing Systems 35 (2022): 26565-26577.
>
>
> Q2: Experiments in Table 1 and Table 2 compare with only EDM training noise schedule. The authors are suggested to compare with more training noise schedules to further verify the effectiveness of training noise schedules.
>
> A2:  We appreciate your suggestion to include comparisons with more training noise schedules to further validate the effectiveness of our approach.
>
> Due to the limited time for rebuttal and our computational resources, we conduct experiments on VP cases to evaluate our WSNR-Equivalent training noise schedule. The results in the following table show that our method further improves performance on VP cases.
>
> | FFHQ | 64 | 128 | 256 |
> | --- | --- | --- | --- |
> | p(WSNR) | 3.80 | **6.23** | **7.99** |
> | p($\sigma$) | 3.80 | 7.41 | 11.78 |
>
>
>
> We understand the importance of ensuring that our paper meets the high standards of ICLR and would greatly appreciate it if you could confirm whether the revisions made align with your expectations and satisfactorily resolve the issues previously highlighted.
>
> Your further guidance and feedback will be important to us and we are committed to making any additional improvements as needed.
>
> Thank you very much for your time and consideration. We look forward to your response.

---

### Official Review · Reviewer_d8X8 · 2023-11-08

**Soundness:** 1 poor
**Presentation:** 1 poor
**Contribution:** 2 fair
**Rating:** 3
**Confidence:** 4

**Summary:**

This paper investigates the noise schedule of diffusion models.
* The authors introduce a training noise schedule according to a metric "weighted signal-to-noise-ratio (WSNR)". It improves FID of latent diffusion models on FFHQ-128/-256 and ImageNet-256.
* The authors propose a sampling noise schedule which slightly improves FID on CIFAR-10 and FFHQ-64 with 35 network evaluations per image.

**Strengths:**

The proposed training noise schedule improves FIDs.

The proposed sampling noise schedule improves FIDs.

**Weaknesses:**

> As illustrated in Fig. 1, we observed substantial disparities in noise levels across images of varying resolutions under the same noise schedule.

Figure 1 has nothing to do with resolutions.

> To the best of our knowledge, we are the first to quantify the noise level of the forward process of the diffusion model, and have successfully extended it to the latent space.

This paper is not the first to quantify the noise level of the forward process of the diffusion model.
* Choi et al., Perception Prioritized Training of Diffusion Models, CVPR2022
* What is the contribution of this paper compared to the above one?

> P·,c(u,v) is the power of the frequency component at (u,v) within the c-th channel.

* Are u and v in the frequency domain?
* What technique is used to convert the images into frequency domain?

Section 4 before 4.1 should be more self-contained.

> Given a finite dataset, an ideal solution for the denoiser D(xt) can be found as the weighted sum of all clean data in the dataset.

* This statement does not have support.
* Eq. 4 describes it but it is not proved.

The proposed method is hardly reproducible.

Writing should be improved. It is hard to follow due to poor connection between consecutive sentences. Especially in Introduction.

Typos:
* > ... in advancing the performance ? diffusion models.
* > Eq. 7 implies that the proportion of data points whose square distance exceeds α times the standard deviation from the mean is? no more than 1/α^2.

Please use one-letter variables in the algorithms for readability.

Please put titles on the axes in the figures for readability.

**Questions:**

This paper proposes a training method and a sampling method. How do they affect the performance when applied together?

What is the number of evaluation points?

How much is the difference in wall clock between 35 network evaluations with the proposed method and typical number of network evaluations with other methods?

Please consult Weaknesses for improving the paper.

---

> ### Author Response · Authors · 2023-11-21
>
> Q1: Figure 1 has nothing to do with resolutions.
>
> A1: Thanks for pointing it out, we acknowledge an error in our initial submission regarding the referencing of figures. Specifically, the discussion of resolutions is visualized in Fig 2 instead of Fig 1. We apologize for any confusion this may have caused and have corrected this reference in our revised manuscript. We appreciate the reviewers bringing this to our attention and ensuring the accuracy of our work.
>
>
>
> Q2: This paper is not the first to quantify the noise level of the forward process of the diffusion model. What is the contribution of this paper compared to [1]?
>
> A2: We have revised the relevant claim in the introduction section. Thanks for your suggestion.
> We note that prior work, P2 [1], proposed a simple heuristic partition of noise levels based on SNR metric:  coarse (SNR 0 ~ 1e^-2), content (1e^-2 ~ 1), clean-up (1 ~ 1e^4).
>
> Unlike our WSNR metric, SNR is neither consistent in different resolutions, nor consistent in latent space. **Our WSNR is the first power spectrum based metric**, **which is consistent for noise level across multiple resolutions in RGB space and latent space**.
>
> However, our quantization metric is different from P2. We summarize the difference as follows:
> |  | Quantitative Metric | Quantitative Method | Consistent on Multiple Resolutions | Consistent in Latent Space |
> | --- | --- | --- | --- | --- |
> | Our | WSNR   | Power spectrum   | Consistent  | Consistent  |
> | P2 [1] | SNR  | Heuristic Discretization  | Not consistent  | Not consistent  |
>
> Besides the WSNR metric, **our paper contributes uniquely by**:
> - We significantly improve the performance of diffusion models, via our novel WSNR-Equivalent training noise scheduler. Specifically, our method decreases the FID score by **3.6** on FFHQ-256x256 dataset, and achieves the **same FID scores** on ImageNet-256 and 512 for UViT-M (#Params: 287M)  as those of UViT-L (#Params: 287M).
> - We analyze that the **ODE sampling noise schedule** should be data-driven. Consequently, we propose estimating the integration interval based on the average data distance.
> - We explore the relationship between the number of evaluation points and generation quality. Our findings lead us to develop a **NFE-guided sampling noise schedule**.
> - Our sampling schedule refines the FID of pre-trained CIFAR-10 and FFHQ-64 models from **1.92 and 2.45 to 1.89 and 2.25**, respectively, utilizing 35 network evaluations per image.
>
> Thus, our paper offers new insights and advancements in the study of the noise schedule of diffusion models.
>
> [1] Choi et al., Perception Prioritized Training of Diffusion Models, CVPR2022

---

> ### Author Response · Authors · 2023-11-21
>
> Q3: Are u and v in the frequency domain?  What technique is used to convert images into frequency domain?
>
> A3:  Yes, u and v are in the frequency domain. Because $P$ represents the power spectrum.
> The power spectrum is defined as
> $$P(u,v) = |F(u,v)|^2 $$
> where $F(u,v)$ is the 2D discrete Fourier transform (DFT). Please check the Eq. (4-67) and (4-89) in [2], which is a textbook widely adopted by university computer vision courses.
>
> [2] Gonzalez, Rafael C. Digital image processing 4th Edition.
>
>
> Q4:  Given a finite dataset, an ideal solution for the denoiser D(xt) can be found as the weighted sum of all clean data in the dataset. The ideal solution is not proved.
>
>
> A4:  We appreciate the reviewer's point regarding the lack of a formal proof for our assertion that an ideal solution for the denoiser can be represented as a weighted sum of all clean data in the dataset.
>
> Here, we provide proof to make it clear:
> Given the standard deviation of Gaussian noise $\sigma$ and the clean data probability distribution $p_{\text{data}}(x) = \frac{1}{Y} \sum_{i=1}^{Y} \delta(x - y_i)$, the probability of the noisy data $p(x; \sigma)$ can be expressed as:
>
> $$
> \begin{align}
> p(x; \sigma) & = p_{\text{data}} * \mathcal{N}(0, \sigma^2 \mathbf{I}) \\
> & = \int_{\mathbb{R}^d} p_{\text{data}}(x_0) \mathcal{N}(x; x_0, \sigma^2 \mathbf{I}) dx_0 \\
> & = \int_{\mathbb{R}^d} \left[ \frac{1}{Y} \sum_{i=1}^{Y} \delta(x_0 - y_i) \right] \mathcal{N}(x; x_0, \sigma^2 \mathbf{I}) dx_0 \\
> & = \frac{1}{Y} \sum_{i=1}^{Y} \int_{\mathbb{R}^d} \mathcal{N}(x; x_0, \sigma^2 \mathbf{I}) \delta(x_0 - y_i) dx_0 \\
> & = \frac{1}{Y} \sum_{i=1}^{Y} \mathcal{N}(x; y_i, \sigma^2 \mathbf{I}) \\
> \end{align}
> $$
>
>
> The loss of the Denoiser is:
>
> $$
> \mathcal{L}(D; \sigma) = E_{y \sim p_\text{data}} \mathbb{E}_{x \sim p(x, \sigma)}  \| D(x; \sigma) - y \|_2^2
> $$
>
> $$\mathcal{L}(D; \sigma) = \int_{\mathbb{R}^d} \frac{1}{Y} \sum_{i=1}^Y \mathcal{N}(x; y_i, \sigma^2 \mathrm{I}) \| D(x; \sigma) - y_i \|_2^2  dx$$
>
> The solution is obtained by setting the gradient w.r.t $D(x; \sigma)$ to zero:
>
> $$
> 0 = \nabla_{D(x;\sigma)} \mathcal{L}(D; x, \sigma)
> $$
>
> $$
> 0 = \nabla_{D(x;\sigma)} \left[ \frac{1}{Y} \sum_{i=1}^Y \mathcal{N}(x; y_i, \sigma^2 \mathrm{I}) \| D(x; \sigma) - y_i \|_2^2 \right]
> $$
>
> $$
> 0 = \sum_{i=1}^Y \mathcal{N}(x; y_i, \sigma^2 \mathrm{I}) \nabla_{D(x;\sigma)} \left[ \| D(x; \sigma) - y_i \|_2^2 \right]
> $$
>
>
> $$
> 0  = \sum_{i=1}^Y \mathcal{N}(x; y_i, \sigma^2 \mathrm{I}) \left[ 2 D(x; \sigma) - 2 y_i \right]
> $$
>
> $$
> 0 = \left[ \sum_{i=1}^Y \mathcal{N}(x; y_i, \sigma^2 \mathrm{I}) D(x; \sigma) \right] - \left[ \sum_{i=1}^Y \mathcal{N}(x; y_i, \sigma^2 \mathrm{I}) y_i \right]
> $$
>
> $$
> D(x; \sigma) = \frac{\sum_{i=1}^Y \mathcal{N}(x; y_i, \sigma^2 \mathrm{I}) y_i}{\sum_{i=1}^Y \mathcal{N}(x; y_i, \sigma^2 \mathrm{I})}
> $$
>
> $$
> D(x; \sigma) = \frac{\sum_{i=1}^Y \mathrm{exp}(\frac{||x-y_i||^2}{-2\sigma^2})  y_i}{\sum_{i=1}^Y \mathrm{exp}(\frac{||x-y_i||^2}{-2\sigma^2})}
> $$
>
> $$
> D(x; \sigma) = \sum_{i=1}^Y \mathrm{softmax}(\frac{||x-y_i||^2}{-2\sigma^2}) y_i
> $$
>
> As shown in the last Equation in this comment, the ideal solution for the denoiser $D(x)$ can be found as the weighted sum of all clean data in the dataset.

---

> ### Author Response · Authors · 2023-11-21
>
> Q5: The proposed method is hardly reproducible.
>
> A5: We will release code to reproduce the results.
>
> Q6: Typos and connection.  Use one-letter variables in algorithms for readability. Put titles on the axes in the figures for readability.
>
> A6: In the revised version of the draft, we fix typos and improve the connection between sentences. Besides, we  make the algorithm and figures more readable. We will keep polishing our draft for the final version. Thanks for your suggestion.
>
> Q7: What is the number of evaluation points?
>
> A7: Evaluation points are the **unique timesteps** at which the diffusion model makes predictions. The 2nd Heun's adopts the start point and end point to estimate the 2nd derivative. Therefore, the end point of the current timestep is identical to the start point of the next timestep. For example, when employing the 2nd Heun method for sampling, which uses 18 timesteps, the Number of Function Evaluations (NFE) is calculated as 2x18 - 1 = 35. Please refer to [this code segment](https://github.com/NVlabs/edm/blob/main/generate.py#L25) for more details.  However, the model actually performs evaluations at only 18 unique timesteps. We refer to these 18 points as evaluation points.  In the case of the midpoint method, all NFEs are performed at unique timesteps, so NFE equals the Number of Evaluation Points (NEP).
>
>
> Q8: How much is the difference in wall clock between 35 network evaluations with the proposed method and typical number of network evaluations with other methods?
>
> A8: There is no difference in wall clock between 35 network evaluations with the proposed method and typical number of network evaluations with other methods.
>
> For a UNet with a base channel of 128 and channels per resolution set to 1-2-2-2, having 4 ResNet blocks at each resolution, the computational cost for each neural network evaluation is **41.72 GFlops** when the output shape is (1, 3, 64, 64). The remaining computational cost involves less than 100 instances of element-wise addition or multiplication operations, amounting to less than **0.0012288 GFlops**.
>
> Therefore, our wall clock time is the same as other methods under the identical NFE.
>
> Q9: This paper proposes a training method and a sampling method.  How do they affect performance when applied together?
>
> A9: To prove the compatibility of our training and sampling methods, we evaluate our trained model on FFHQ-256 using our proposed sampling method. The table shows that our sampling method consistently lowers the FID score of the trained model, indicating improved performance since a lower FID score is desirable.
>
> |  | Heun (NFE=35) | Ours (NFE=35) |
> | --- | --- | --- |
> | p(WSNR) | 7.89 | **7.40** |
> | p($\sigma$) | 11.49 | **11.02** |
>
> We understand the importance of ensuring that our paper meets the high standards of ICLR and would greatly appreciate it if you could confirm whether the revisions made align with your expectations and satisfactorily resolve the issues previously highlighted.
>
> Your further guidance and feedback will be important to us and we are committed to making any additional improvements as needed.
>
> Thank you very much for your time and consideration. We look forward to your response.

---

> ### Author Response · Authors · 2023-11-22
> **Follow-up**
>
> Dear Reviewer d8X8,
>
> We appreciate the diligent efforts put into reviewing our work.
>
> Like you, we believe that the writing improvement and readability are important.
>
> We have revised the manuscript and would greatly appreciate it if you could confirm whether the revisions made align with your expectations and satisfactorily resolve the issues previously highlighted.
>
> We would happily address any other questions.
>
> Thank you,
> The authors

---

> ### Author Response · Authors · 2023-11-23
>
> Dear Reviewer d8X8,
>
> Just a reminder, there are only 4 hours left for the rebuttal.
>
> After this time, we will not be able to participate in the discussion any further.
>
> We are waiting for your response. We would happily address any other questions.
>
> Best Regards,
>
> The Authors

---

> > ### Author Response · Authors · 2023-11-23
> >
> > Dear Reviewer d8X8,
> >
> > Just a reminder, there are only **3 hours** left for the rebuttal.
> >
> > After this time, we will not be able to participate in the discussion any further.
> >
> > We are waiting for your response. We would happily address any other questions.
> >
> > Best Regards,
> >
> > The Authors

---

> ### Author Response · Authors · 2023-11-23
>
> Dear Reviewer d8X8,
>
> Just a reminder, there are **only 2 hours** left for the rebuttal.
>
> After this time, we will not be able to participate in the discussion any further.
>
> We are waiting for your response. We would happily address any other questions.
>
> Best Regards,
>
> The Authors

---

### Author Response · Authors · 2023-11-22
**Awaiting Feedback**

Dear Reviewers,

We sincerely thank you for the time and effort dedicated to reviewing our work. Your insights and suggestions are valuable to us, and we deeply appreciate your contribution to the peer-review process.


As the rebuttal discussion period is drawing to a close **tomorrow**, we are eagerly awaiting any feedback, questions, or comments you may have. Your perspectives are crucial for us to gain a deeper understanding of the review and to guide any necessary improvements in our submission.

We understand that you have a busy schedule, and reviewing papers is a time-consuming task. Therefore, we greatly appreciate the effort you put into this process. If there are any specific queries or additional information required from our side, please feel free to let us know, and we will be more than happy to provide it.


Warm regards,

Authors

---

### Author Response · Authors · 2023-11-23

Dear Reviewers,

Due to the absence of a second-stage discussion phase in this year's ICLR review process, it means that we **won't be able to engage in further discussions with the reviewers** after **9 hours**.

We sincerely hope to receive a response from the reviewers.

If there are any specific queries or additional information required, we will be happy to provide it.

Best Regards,

Authors

---

### Meta-Review · Area_Chair_aR8r · 2023-12-11

**Metareview:**

**Summary**

This paper investigate the noise schedule of diffusion models. For training, the authors propose the usage of weighted signal-to-noise ratio (WSNR) as the guidance for training diffusion models with different resolutions. For sampling, the paper proposes an adaptive data-driven sampling scheme for balancing efficacy and efficiency,


**Strength**

1. Both the proposed training and sampling strategies for noise schedules improve the overall FIDs on two sets of benchmarks.

2. The paper proposes a data-driven sampling schedule to ensure the diversity of generations.


**Weaknesses**

1. The position of the paper is very unclear. It seems the paper put several scattered irrelevant pieces into a single paper where the only connection between each discussed point is “noise schedule.” Even the evaluation sets for the training/sampling schedule are different. This actually makes it hard to draw major contributions from one paper.

2. Using WSNR to improve the training of different resolution, and even the latent spaces, is not very novel, which diminish one of the main
 claims of the paper.

3. The initial version lacks enough comparisons to combine the proposed methods. While the authors include additional numbers, the evaluation is not complete. Also, experiments with resolutions higher than 256 should also be part of the evaluation.

**Justification For Why Not Higher Score:**

I generally agree with the points raised by d8X8, QVZ8, and these issues are hard to address without a significant content change.

1. As mentioned by the authors' response, it is fine and even better to improve the training and inference strategies of diffusion models simultaneously. However, a clear and coherent storyline should be discussed and well-motivated. For example, targeting the goal of high-resolution or fewer NFEs. The current form of the paper is not properly motivated.

2. Modifying SNR to support diffusion models on variant resolutions is also not new after all. AFAIK, f-DM [1] should be one of the first papers to discuss how one should change the definition of SNR when working on different resolutions, including latent spaces. Later, both [2] [3] proposed heuristics to adjust the noise schedule, which essentially falls back to a similar formulation as f-DM proposed.

[1] f-DM:	A multi-stage diffusion model via progressive signal transformation, ICLR 2023

[2] simple diffusion: End-to-end diffusion for high-resolution images, ICML 2023

[3] On the importance of noise scheduling for diffusion models, arXiv 2023


3. While the adaptive sampling scheme based on NFE to choose different solver methods is new (which can be an important point to explore), the initial analysis of the reverse ODE view and the ideal output of denoiser has already been partially discussed in EDM. Also, it is hard to draw a connection between Eq 8 to the NFE-guided approach.

**Justification For Why Not Lower Score:**

N/A

---

### Decision · Program_Chairs · 2024-01-16

Reject